# Academic resilience in European countries: The role of teachers, families, and student profiles

**Francisco J. García-Crespo[1], Rubén Fernández-Alonso [2]\*, José Muñiz[3]**

**1** Universidad Complutense de Madrid, Madrid, Spain, **2** Universidad de Oviedo, Oviedo, Principado de Asturias, Spain, **3** Universidad de Nebrija (Nebrija University), Madrid, Spain

\* fernandezaruben@uniovi.es

## Abstract

Academic resilience is a student's ability to achieve academic results significantly higher than would be expected according to their socioeconomic level. In this study, we aimed to identify the characteristics of students, families, and teacher activities which had the greatest impact on academic resilience. The sample comprised 117,539 fourth grade students and 6,222 teachers from 4,324 schools in member states of the European Union that participated in the PIRLS 2016 study. We specified a two-level hierarchical linear model in two phases: in the first level we used the students' personal and family background variables, in the second level we used the variables related to teaching activity. In the first phase we used the complete model for all countries and regions, in the second phase we produced a model for each country with the highest possible number of statistically significant variables. The results indicated that the students' personal and family variables that best predicted resilience were the reading self-confidence index, which increased the probability of student resilience by between 62 and 130 percentage points, a feeling of belonging to the school, which increased the chances of being resilient by up to 40 percentage points, and support from the family before starting primary school (Students from Lithuania who had done early literary activities in the family setting were twice as likely to be resilient than those who had not). The teaching-related factors best predicting resilience were keeping order in the classroom, a safe and orderly school environment (increasing chances of resilience by up to 62 percentage points), and teaching focused on comprehension and reflection, which could increase the probability of resilience by up to 61 percentage points.

## Introduction

Evaluations of education systems are a key tool for describing the development of students' skills and how schools help in improving learning [1]. Identifying the contextual factors of the teaching process that have the greatest impact on academic performance may help to prevent school failure, and may also help to guide policy decisions towards the continual improvement of the education system [2–5]. Because of their usefulness, more and more countries are participating in comparative studies of education systems run by international organizations such as

**Data Availability Statement:** The International Databases of PIRLS 2016 can be accessed from this link: https://timssandpirls.bc.edu/pirls2016/international-database/index.html. From it, two data files are downloaded in SPSS format (also

available for Stata): P16_SPSSData_pt1.zip (AAD-GEO 100MB) and P16_SPSSData_pt2.zip (HKG-ZA5 100MB). The data files are public and without access restrictions. It is also possible to download the supporting documents (User Guide and Supplements files) to replicate the analyzes.

**Funding:** This research was funded by the Spanish Ministry of Economy, Industry and Competitiveness, Reference PSI2017-85724-P and by Universidad de Oviedo (Spain). Reference FUO-18-262.

**Competing interests:** The authors have declared that no competing interests exist.

the Organization for Economic Co-operation and Development (OECD), the International Association for the Evaluation of Educational Achievement (IEA), and the European Commission (EC), among others [6]. In this context, the study of academic resilience is a thriving area of growing interest in current educational research [7–13].

Academically resilient students are those who achieve academic success despite adverse socioeconomic conditions [14, 15]. As Choi & Calero [16] noted, students' capacity for resilience comes from the interaction between personal, family, and school variables. Academic resilience is closely connected to the student's personality characteristics such as socio-affective variables, self-concept, academic expectations, causal attributions, and confidence in their own abilities [17–22]. There is also a broad consensus among researchers that academic resilience is strongly linked to motivational variables such as effort, persistence, personal strength, the ability to work autonomously, enthusiasm for learning, and enjoyment of reading [19–21, 23–25].

The family context also seems to be associated with the likelihood of being academically resilient, and parents' academic expectations have been shown to be a key predictor of educational results [26]. Fernández-Alonso, Álvarez-Díaz, Woitschach, Suárez-Álvarez, & Cuesta [27] found that students whose parents had a more distant or indirect profile of family involvement tended to demonstrate better results than students from homes with more controlling styles. Using the TIMSS 2011 database, Sandoval & Bialowolski [28] found that high academic expectations and time spent on mathematics in the home had a positive effect on Singaporean students. In a sample of Chinese students participating in university entrance exams, Li [29] found that supervision by parents and school involvement and recognition strengthened resilience. Studies such as Cheung et al. [30], reported that students with family support tended to have better psychological wellbeing, and were more likely to be resilient.

Educational research has accumulated evidence relating the school learning context to academic resilience. Erberber et al. [17], found that the school factors most strongly related to resilience in mathematics and science subjects included teacher expectations of student performance, the school's interest in academic success, a safe school atmosphere, school discipline, and the amount of educational resources available. On similar lines García-Crespo et al. [14], found that a favorable school environment notably increased student academic resilience. Another line of research that has produced very consistent results indicates teachers as a key factor in academic resilience [31–35]. Although the concept of teaching quality is multidimensional [36–39], researchers have been able to summarize a set of teaching practices and didactic strategies with real potential for improving student motivation, and improving their learning outcomes [33, 40–47]. Variables such as teachers satisfaction with their work also play an important role in their performance, and consequently in the students' academic performance [48–50]. In addition, better quality education systems are able to attract highly trained, skilled teachers, offering them careers which recognize and enable teacher development and training [51–53], which suggests that initial and continued teacher training would also be associated with students' academic resilience. Consequently, there is great political interest in assessing whether participation in continual professional development activities is driving changes in teaching practices and in student performance, and whether some types of activities are more effective than others [33, 54–56]. Research has found that participation in professional development activities is linked to individual motivation and the desire to improve teaching skills in order to be able to help students [57]. This participation has a direct positive impact on improving student performance [56], and increasing the likelihood of resilience closing the gap between students [58].

Within this context, the main objective of this current study is to identify and assess the influence of two types of variables on students' academic resilience: students' personal and family variables, and the teachers' teaching practices.

## Materials and methods

### Participants

We defined the target population as students in the 4[th] year of compulsory education in the European Union countries and regions taking part in PIRLS 2016. PIRLS is designed to describe and summarize student performance, which is why it aims for the target population to have complete coverage. However, in some cases, for political, geographical, or operational reasons, complete national coverage is not achieved. For this reason, in some exceptional situations, they permit schools to be excluded (inaccessibility due to a geographically remote location, extremely small size, offering a radically different grade structure or curriculum to the mainstream educational system or providing instruction solely to students in the student-level exclusion categories below) or the exclusion of students within schools (students with functional or intellectual disabilities) [59]. Within each country, the sampling, which was in accordance with international test standards [59], was stratified, sequential by cluster, and two-stage. In the first stage, schools were selected with a probability proportional to their size within each stratum. In the second stage the class or classes to participate within the school were selected. The sample was made up of 117,539 students and 6,222 teachers from 4,324 schools (Table 1). Table 1 gives a description of the sample. In this study, we used 23 samples

**Table 1. Sample description.**

| Country | Number of Students | Number of Teachers | Number of Schools | Number of home questionnaires | Coverage of the national target population | Coverage of home |
|---|---|---|---|---|---|---|
| Austria | 4,360 | 257 | 150 | 4,074 | 94.4% | 94.2% |
| Belgium (Flemish) | 5,198 | 277 | 148 | 4,560 | 98.4% | 87.6% |
| Belgium (French) | 4,623 | 254 | 158 | 3,971 | 94.0% | 85.6% |
| Bulgaria | 4,281 | 214 | 153 | 4,206 | 95.7% | 97.8% |
| Czech Republic | 5,537 | 269 | 157 | 5,202 | 96.6% | 94.4% |
| Denmark | 3,508 | 186 | 185 | 3,214 | 91.2% | 91.0% |
| England | 5,095 | 210 | 170 | 0 | 96.3% | 0.0% |
| Finland | 4,896 | 295 | 151 | 4,535 | 97.6% | 93.2% |
| France | 4,767 | 284 | 163 | 4,218 | 94.6% | 89.6% |
| Germany | 3,959 | 221 | 208 | 2,668 | 95.8% | 66.8% |
| Hungary | 4,623 | 209 | 149 | 4,374 | 95.5% | 94.4% |
| Ireland | 4,607 | 219 | 148 | 4,254 | 96.9% | 92.3% |
| Italy | 3,940 | 217 | 149 | 3,586 | 95.1% | 91.3% |
| Latvia | 4,157 | 218 | 150 | 3,882 | 92.1% | 93.4% |
| Lithuania | 4,317 | 243 | 195 | 3,623 | 95.8% | 86.4% |
| Malta | 3,647 | 207 | 95 | 3,155 | 92.1% | 86.5% |
| Netherlands | 4,206 | 226 | 132 | 2,246 | 96.9% | 53.4% |
| Northern Ireland | 3,693 | 160 | 134 | 1,445 | 97.0% | 37.7% |
| Poland | 4,413 | 246 | 148 | 4,290 | 96.1% | 97.4% |
| Portugal | 4,642 | 318 | 218 | 4,514 | 92.5% | 97.4% |
| Slovak Republic | 5,451 | 334 | 220 | 5,210 | 95.2% | 95.7% |
| Slovenia | 4,499 | 253 | 160 | 4,256 | 97.6% | 95.1% |
| Spain | 14,595 | 678 | 629 | 13,402 | 95.2% | 91.1% |
| Sweden | 4,525 | 227 | 154 | 3,758 | 94.8% | 84.8% |
| Total | 117,539 | 6,222 | 4,324 | 98,643 | | |

from 22 EU countries, as Belgium has two samples, one Flemish-speaking and the other French-speaking. England was not included in the analysis as they did not provide data for the family questionnaire, which prevented the creation of student socioeconomic and sociocultural indices, something that was essential for identifying resilient students.

## Procedure

The PIRLS 2016 test was applied following the standards outlined by the International Association for the Evaluation of Educational Achievement (IEA) [59]. The test was applied on a single day, structured in two 40-minute sessions with a 30-minute break in the middle. Each session involved reading a literary text and an informative text (not necessarily in that order) and answering a series of items about them. The test was specified according to the theoretical framework established in Mullis and Martin [60], and comprised a total of 12 readings (half informative, half literary) distributed in 16 different test booklets following a partially balanced incomplete block design [61]. Because each student only completed a single test booklet, when the test was given, test-booklets were distributed so that each item was answered by a similar number of students. Once the reading sessions were finished, the students completed a background questionnaire (Student Questionnaire) which would be used to complement the information about student reading comprehension. The process also included a Home Questionnaire (Learning to Read Survey) for families, a Teacher Questionnaire which was completed by the language teacher, and a School Questionnaire which was completed by the heads of the schools. It should be noted that due to the data collection procedure for the Home Questionnaire, it was not available for all of the students who participated in the study because some families did not return a completed questionnaire. Regardless of this, the number of questionnaires available was sufficient to perform the analysis for this study as it achieved sufficient coverage with the application of missing data recovery techniques detailed below.

## Measurement instruments and variables

**Academic resilience.**　A student is considered to show resilience if they meet two conditions: a) their score in the *Index of Economic, Social and Cultural Status (ESCS)* is in the lowest quarter of the ESCS in their country, and b) their score in the PIRLS 2016 *Reading Comprehension* is higher than the third quartile of overall achievement once the individual ESCS is discounted. The full method of calculation may be found in García-Crespo et al. [14].

The ESCS index is essentially unidimensional [62–65] and is constructed from four items in the student context and family questionnaires: home possessions, books at home, parents' highest education level, and parents' highest occupation level.

The score in *Reading Comprehension* was calculated from the responses to the cognitive reading tests and information in the background questionnaires. It was constructed by applying models derived from Item Response Theory (IRT), assigning five plausible values as scores on a scale with a mean of 500 and standard deviation of 100 [59].

**Predictor variables of academic resilience.**　We considered 24 variables to predict academic resilience, eight related to students, two to families, and fourteen to teaching practices.

**Student-related variables.**　The variables *Gender* and *Attended a preschool educational program* (less than 2 years; two years or more) are dichotomous, the remaining variables were constructed using IRT partial credit scaling [59, 66]. For this analysis they were normalized with a mean of 0 and standard deviation of 1 *[N(0,1)]*. The six remaining variables (and their labels in brackets) in the student questionnaire were as follows:

*Sense of school belonging (Sensebel)*. Students were asked how much they agreed with five statements about their attitude toward school, 1 (I like being in school), 2 (I feel safe when I

am at school), 3 (I feel like I belong at this school), 4 (Teachers at my school are fair to me), and 5 (I am proud to go to this school).

*Engaged in reading lessons (Engagedr)*. Students were scored according to their degree of agreement with nine statements related to their reading commitment: 1 (I like what I read about school), 2 (My teacher gives me interesting things to read), 3 (I know what my teacher expects me to do), 4 (My teacher is easy to understand), 5 (I am interested in what my teacher says), 6 (My teacher encourages me to say what I think about what I have read), 7 (My teacher lets me show what I have learned), 8 (My teacher does a variety of things to help us learn), and 9 (My teacher tells me how to do better when I make a mistake).

*Like reading (Likeread)*. Students were scored on this scale according to their degree of agreement with eight statements and how often they did two reading activities outside of school: 1 (I like talking about what I read with other people), 2 (I would be happy if someone gave me a book as a present), 3 (I think reading is boring), 4 (I would like to have more time for reading), 5 (I enjoy reading), 6 (I learn a lot from reading), 7 (I like to read things that make me think), and 8 (I like it when a book helps me imagine other worlds).

*Confident in reading (Confiden)*. Students were scored according to their degree of agreement with six statements: 1 (I usually do well in reading), 2 (Reading is easy for me), 3 (I have trouble reading stories with difficult words), 4 (Reading is harder for me than for many of my classmates), 5 (Reading is harder for me than any other subject), and 6 (I am just not good at reading).

*Early literacy activities before school (Litactiv)*. Students were scored according to how often their parents' did the nine activities: 1 (Read books), 2 (Tell stories), 3 (Sing songs), 4 (Play with alphabet toys, e.g., blocks with letters of the alphabet), 5 (Talk about things you had done), . . .

*Early literacy tasks (Littask)*. Students were scored according to their parents' responses about how well their children could do the six tasks: 1 (Recognize most of the letters of the alphabet), 2 (Read some words), 3 (Read sentences), 4 (Read a story), 5 (Write letters of the alphabet), and 6 (Write some words).

**Family-related variables.** We extracted two variables from the family questionnaire, constructed and scored in the same way [$N(0,1)$] as the variables from the student questionnaire.

*Parents' perceptions of child's school (Parentsp)*. Students were scored on this scale according to their parents' responses to six statements about the school: 1 (My child's school does a good job including me in my child's education), 2 (My child's school provides a safe environment), 3 (My child's school cares about my child's progress in school), 4 (My child's school does a good job informing me of their progress), 5 (My child's school promotes high academic standards), and 6 (My child's school does a good job in helping them become better in reading).

*Parents like reading (Parentsl)*. Students were scored on this scale according to their parents' responses to eight statements about reading and how often they read for enjoyment: 1 (I read only if I have to), 2 (I like talking about what I read with other people), 3 (I like to spend my spare time reading), 4 (I read only if I need information), 5 (Reading is an important activity in my home), 6 (I would like to have more time for reading), 7 (I enjoy reading), and 8 (Reading is one of my favorite hobbies)

**Teaching-related variables.** We extracted 14 variables from the teacher questionnaire. Gender was dichotomous, the others were expressed on a continuous, normalized scale [$N(0,1)$].

*Teachers' basic training (Basictraining)*. Students were scored on this scale according to their teachers' responses (Not at all / Overview or introduction to topic / It was an area of emphasis) to four statements about their formal education and training and the extent to

which they studied the following areas: 1 (language of test), 2 (Literature), 3 (Pedagogy/teaching Reading), and 4 (Educational psychology).

*Teachers' complimentary training (Complime).* Students were scored on this scale according to their teachers' responses (Not at all / Overview or introduction to topic / It was an area of emphasis) to three statements about their formal education and training regarding the extent to which they studied the following areas: 1 (Remedial reading), 2 (Reading theory), and 3 (Assessment methods in reading).

*School emphasis on academic success (Emphasis).* Students were scored according to their teachers' responses characterizing twelve aspects of the School Emphasis on Academic Success scale: 1 (Teachers' understanding of the school's curricular goals), 2 (Teachers' degree of success in implementing the school's curriculum), 3 (Teachers' expectations for student achievement), 4 (Teachers' ability to inspire students), 5 (Collaboration between school leadership (including master teachers) and teachers for planning instruction), 6 (Parental involvement in school activities), 7 (Parental commitment to ensure that students are ready to learn), 8 (Parental expectations for student achievement), 9 (Parental support for student achievement), 10 (Students' desire to do well in school), 11 (Students' ability to reach the school's academic goals), and 12 (Students' respect for classmates who excel academically).

*Safe and orderly school (Security).* Students were scored according to their teachers' degree of agreement with eight statements on the Safe and Orderly School scale: 1 (This school is located in a safe neighborhood), 2 (I feel safe at this school), 3 (This school's security policies and practices are sufficient), 4 (The students are well behaved), 5 (The students are respectful of the teachers), 6 (The students respect school property), 7 (This school has clear rules about student conduct), and 8 (This school's rules are fairly and consistently enforced)

*Teacher interaction (Interact).* Students were scored on this scale according to their teachers' responses to four statements about different types of interaction with other teachers in terms of how often they occurred: 1 (Share what I have learned about my teaching experiences), 2 (Observe another classroom to learn more about teaching), 3 (Work together to improve how to teach a particular topic), 4 (Work with teachers from other schools on the curriculum), and 5 (Work with teachers from other grades to ensure continuity in learning).

*Teacher job satisfaction (Satisfac).* Students were scored according to how often their teachers responded positively to the five statements on the Teacher Job Satisfaction scale: 1 (I am content with my profession as a teacher), 2 (I find my work full of meaning and purpose), 3 (I am enthusiastic about my job), 4 (My work inspires me), and 5 (I am proud of the work I do).

*Classroom instruction limited by student attributes (Limitat).* Students were scored according to their teachers' responses about seven attributes of their students that could limit how they teach their class in the Classroom Instruction Limited by Student Attributes scale: 1 (Students lacking prerequisite knowledge or skills), 2 (Students suffering from lack of basic nutrition), 3 (Students suffering from not enough sleep), 4 (Students absent from class), 5 (Disruptive students), 6 (Uninterested students), 7 (Students with mental, emotional, or psychological impairment), and 8 (Lack of support for using information technology)

*Routine strategies for reading (Routinare).* Students were scored on this scale according to their teachers' responses to three statements about reading activities regarding how often they did them: 1 (Read aloud to students), 2 (Ask students to read aloud), and 3 (Ask students to read silently on their own).

*Systematic strategies for reading (Sistemat).* Students were scored on this scale according to their teachers' responses to four statements about reading activities regarding how often they did them: 1 (Teach students strategies for decoding sounds and words), 2 (Teach students new vocabulary systematically), 3 (Teach students how to summarize the main ideas), and 4 (Teach or model skimming or scanning strategies).

*Use of comprehension reading techniques (Comprehe)*. Students were scored on this scale according to their teachers' responses to three statements about how often they did things to help develop reading comprehension skills: 1 (Locate information within the text), 2 (Identify the main ideas of what they have read), and 3 (Explain or support their understanding of what they have read).

*Use of reflective reading techniques (Reflecti)*. Students were scored on this scale according to their teachers' responses to six statements about how often they did things to help develop reading strategies: 1 (Compare what they have read with experiences they have had), 2 (Compare what they have read with other things they have read), 3 (Make predictions about what will happen next in the text they are reading), 4 (Make generalizations and draw inferences based on what they have read), 5 (Describe the style or structure of the text they have read), and 6 (Determine the author's perspective or intention).

*Homework tracking (Homework)*. Students were scored on this scale according to their teachers' responses to three statements about how often they did the following with the reading homework assignments for their class: 1 (Correct assignments and give feedback to students), 2 (Discuss the homework in class), and 3 (Check whether the homework was completed).

*Selection of adapted readings (Readings)*. Students were scored on this scale according to their teachers' responses to three statements about how often they did the following in teaching reading to their class: 1 (Provide reading materials that match the students' interests), 2 (Provide materials that are appropriate for the reading levels of individual students), and 3 (Give students time to read books of their own choosing).

The variables *Emphasis*, *Security*, *Satisfact*, and *Limitati* were constructed using the IRT partial credit scaling model [59, 66], while the remaining variables were constructed via Principal Component Analysis (PCA)

The TIMSS & PIRLS International Study Center thoroughly reviewed the items to assess and evaluate their psychometric characteristics. This review allowed them to detect items with unusual properties that could indicate problems or errors for a particular country. Countries are expected to exhibit some variation in the item responses, however, when that variation is large there is said to be item-country interaction, and measures need to be taken to resolve the problem. To detect these interactions, the TIMSS & PIRLS International Study Center produced a graphical display for each item showing the difference between the Rasch difficulty of a parameter for an item in a country and for the item in the international average. In each of these item-by-country interaction displays, the difference in the Rasch difficulty for each country is presented as a 95% confidence interval, including a Bonferroni correction for multiple comparisons between participating countries [59].

## Data analysis

We used multilevel logistic regression models to analyze the influence of the predictor variables on the criterion (academic resilience) [67–69]. The use of hierarchical linear models is due to the structure of the data matrices in the international evaluations of education systems. Analysis procedures derived from the classical general linear model assume that the cases are selected via simple random sampling, however, in large scale educational evaluations, the assumption of independence of collected data is not usually met [70]. In fact, PIRLS 2016 used a complex sampling design, in which the observations are definitely not independent, as the students (level 1) within the same class or school (level 2) are more similar to each other than to the students in other classes or schools [71]. In added designs, each level of the hierarchy has a different variability and the errors are not independent. Because the analytical procedures derived from the classical general linear model do not consider this interdependence of

cases, their results may very well be biased due to underestimating standard errors which may cause false significance. Multilevel models, by addressing this grouped sample design, are a valid alternative to the replicated weightings offered initially by the PIRLS 2016 database. For each of the countries analyzed, we specified two multilevel logistical regression models. In the first iteration, the model included all of the predictor variables. It followed a non-centered model, as the second-level coefficients provide correct estimations of individual effects and the contextual effect when the contextual predictor variable is included in the second level of the model [72]. In the second iteration, the model was specified which retained the predictors that were statistically significant in the initial model for each country. This means that we produced as many models as participating countries. That will allow a comparison of which variables were significant in within each country and to what extent.

We considered the following parameters to analyze and evaluate the models obtained:

a.  Coefficients ($\beta_i$, i = 1, . . ., 10; $\gamma_{0j}$, j = 1, . . ., 14) and their signs. Positive values would indicate direct positive impact of the predictor variable on the criterion variable, negative values would indicate an inverse impact.

b.  P-value of the coefficients ($\beta_i$, i = 1, . . ., 10; $\gamma_{0j}$, j = 1, . . ., 14): marginal level of significance. We selected the variables that were significant at 10% and at 5%.

c.  Odds ratio ($e^{\beta_i}$, $e^{\gamma_{0j}}$). This allows comparison of the odds of different values of a $\beta_i$ or $\gamma_{0j}$ predictor variable, indicating the amount of impact, with a value of 1 indicating that $\beta_i$ or $\gamma_{0j}$ have no impact. The further from 1, the greater the impact, whether direct or inverse (García-Crespo et al., 2019).

To specify the models, we used HLM6$^{©}$ software and cases were weighted by the original weightings of students and schools in the PIRLS 2016 database. These weightings, which reflect the probabilities of selecting students and schools in the study, allow proper reproduction of the population size and optimize the representativeness of the results [73]. The response rate of the questionnaires from which the indices were constructed was very high, over 95% for the student, teacher, and school head questionnaires and over 85% for the family questionnaire. There were no concentrations of missing data by country or school, which means it did not bias the responses. Due to the construction of the indices, they did not give anomalous data, as all of the continuous indices were from standardized distributions with a mean of 0 and a standard deviation of 1. Although there are many methods for recovering missing data [74], in this study we used the linear trend at point process in the Missing Value Analysis module in SPSS, using the class the student belonged to as segmentation.

## Results

Table 2 shows the ESCS and the percentages of academically resilient students in reading by country, along with the standard errors of the estimated parameters.

The countries with the highest proportions of resilient students, according to the estimations produced from the definition of resilience used in this study, were Poland (42.22%) and Italy (40.57%). The French-speaking area of Belgium (5.96%) and Malta (6.45%) had the lowest proportions. This indicates a great variation in the proportions of resilient students between the different countries analyzed.

Before presenting the results of this study, the bilateral correlations between the variables in the model are provided, in order to reject excessive correlation that would prevent the true impact of each variable from being seen independently. Table 3 shows the bivariate correlations between the student and family contextual variables, while Table 4 shows the correlations

**Table 2. Index of economic, social and cultural status, and percentage of resilient students for the European Union countries.**

| Country | ESCS | ESCS s.e. | Resilient percentage | Resiliente percentage s.e. |
|---|---|---|---|---|
| Austria | 0.09 | 0.03 | 16.43 | 1.80 |
| Belgium (Flemish) | 0.25 | 0.03 | 12.80 | 1.25 |
| Belgium (French) | 0.11 | 0.03 | 5.96 | 0.77 |
| Bulgaria | -0.24 | 0.06 | 27.48 | 3.40 |
| Czech Republic | 0.09 | 0.03 | 22.88 | 1.81 |
| Denmark | 0.65 | 0.03 | 17.45 | 1.56 |
| Finland | 0.49 | 0.02 | 32.76 | 1.95 |
| France | 0.01 | 0.03 | 11.88 | 1.21 |
| Germany | 0.01 | 0.04 | 20.00 | 1.86 |
| Hungary | 0.00 | 0.06 | 25.33 | 2.15 |
| Ireland | 0.23 | 0.03 | 36.23 | 1.90 |
| Italy | -0.45 | 0.04 | 40.57 | 1.88 |
| Latvia | 0.30 | 0.03 | 27.18 | 1.77 |
| Lithuania | 0.11 | 0.03 | 23.14 | 1.76 |
| Malta | -0.10 | 0.01 | 6.45 | 0.77 |
| Netherlands | 0.45 | 0.03 | 19.23 | 1.85 |
| Northern Ireland | 0.34 | 0.03 | 36.90 | 2.03 |
| Poland | -0.04 | 0.03 | 42.22 | 2.20 |
| Portugal | -0.20 | 0.03 | 24.95 | 1.80 |
| Slovak Republic | -0.16 | 0.04 | 18.92 | 1.71 |
| Slovenia | 0.16 | 0.03 | 22.09 | 1.90 |
| Spain | -0.02 | 0.03 | 20.93 | 1.10 |
| Sweden | 0.66 | 0.03 | 18.80 | 1.61 |

s.e.: Standard error

between the teaching process variables. Although the correlation was statistically significant in all cases, the values, as shown below were almost all low.

The tables below provide the results of the models that only conserved the statistically significant variables to the resilience condition by country, models from the second iteration. Table 5 gives the contextual variables from the students and their families, and Table 6 gives the teaching context variables. Both tables contain the parameters of the logistical regression by country and dependent variable: the coefficients and their standard errors, the odds ratio, and the p-value (level of significance).

**Table 3. Correlations of student and family contextual variables.**

| | SenseBelonging | EngagedReading | LikeReading | Confident | LitActivities | LitTask | ParentsPercep | ParentsLikeRead |
|---|---|---|---|---|---|---|---|---|
| SenseBelonging | 1 | .580** | .428** | .103** | .051** | .088** | .183** | .044** |
| EngagedReading | | 1 | .527** | .139** | .057** | .075** | .167** | .032** |
| LikeReading | | | 1 | .171** | .080** | .135** | .114** | .089** |
| Confident | | | | 1 | .106** | .139** | .022** | .074** |
| LitActivities | | | | | 1 | .290** | .135** | .314** |
| LitTask | | | | | | 1 | .143** | .164** |
| ParentsPercep | | | | | | | 1 | .101** |
| ParentsLikeRead | | | | | | | | 1 |

** The correlation is significant at the 0.01 level (bilateral)

**Table 4. Correlations of variables related to teaching.**

| | BasicTraining | CompleTraining | Emphasis | Security | Interaction | Satisfaction | Limitations | Rutinare | Sistemat | Comprehensive | Reflective | Homework | Readings |
|---|---|---|---|---|---|---|---|---|---|---|---|---|---|
| BasicTraining | 1 | .448** | .103** | .131** | .119** | .069** | -.035** | .072** | .099** | .074** | .104** | .042** | .104** |
| CompleTraining | | 1 | .092** | .064** | .139** | .120** | .035** | .021** | .155** | .070** | .157** | .077** | .082** |
| Emphasis | | | 1 | .554** | .380** | .345** | .342** | .123** | .146** | .075** | .164** | .057** | .209** |
| Security | | | | 1 | .260** | .301** | .384** | .053** | .058** | .011** | .086** | .012** | .166** |
| Interaction | | | | | 1 | .339** | .139** | .179** | .356** | .248** | .344** | .192** | .224** |
| Satisfaction | | | | | | 1 | .238** | .148** | .227** | .144** | .212** | .099** | .223** |
| Limitations | | | | | | | 1 | .005* | .067** | .013** | .087** | .066** | .019** |
| Rutinare | | | | | | | | 1 | .376** | .369** | .306** | .163** | .246** |
| Sistemat | | | | | | | | | 1 | .523** | .627** | .346** | .324** |
| Comprehensive | | | | | | | | | | 1 | .591** | .275** | .273** |
| Reflective | | | | | | | | | | | 1 | .295** | .323** |
| Homework | | | | | | | | | | | | 1 | .089** |
| Readings | | | | | | | | | | | | | 1 |

** The correlation is significant at the 0.01 level (bilateral).

* The correlation is significant at the 0.05 level (bilateral)

Tables 7 and 8 show the percentages of countries in which the coefficient of the corresponding variable was statistically significant.

The first notable result is that the teaching context variables had less impact on the probability of resilience than the student context variables. In this regard, in every country, students with greater confidence in reading tended to be more likely to be resilient that those who did not. It is also worth highlighting that in around half of the countries, the sense of belonging to the school, positive attitudes in reading classes, liking reading, and having done reading tasks in the family setting had a positive impact on the capacity for resilience. Teaching in secure settings, having had complementary training, and greater emphasis on academic results were the aspects of the teaching context that increased the chances of students being resilient in the greatest number of countries.

## Discussion and conclusions

The main objective of our study was to analyze the extent to which students' personal and family characteristics and their teachers' teaching activities were linked to the students' academic resilience. Students show academic resilience when, despite unfavorable socioeconomic and sociocultural levels, they have good academic performance. Our results show that the weight of the predictor variables in academic resilience varied considerably between countries, which may be, as Mullis, Martin, Goh, & Prendergast [75], noted, due to social and cultural differences, and the differences between different countries' education systems. The student-related variable that was most strongly linked to academic resilience was *confidence in reading*, which was statistically significant and positive in all countries, a finding that is in line with data from Martin & Marsh [19], Vaknin-Nusbaum, et al. [20], Veas, et al. [21], and Wosman, et al. [22]. Nonetheless, it is worth noting that studies such as Marsh & Craven [76], via research based on a reciprocal effects model and a meta-analysis showed that prior academic self-concept (rather than self-esteem) and prior success had positive effects on subsequent self-concept and subsequent success. The next most strongly linked variable was the *students' sense of school belonging*, which also agrees with previous results [17]. Other student personal characteristics were statistically significant in a good number of countries, such as being a boy or girl, or having attended pre-primary school. Students being engaged in reading lessons demonstrated

**Table 5. Coefficient, p-value, and odds ratio of student and family contextual variables.**

| | | Gender | Preprimary | Sensebelonging | Engagedreading | Likereading | Confident | LitActivities | LitTask | ParentsPercep | ParentsLikeRead |
|---|---|---|---|---|---|---|---|---|---|---|---|
| Austria | Coefficient | | | | -0.27 | | 0.83 | 0.22 | | | |
| | Coeff_s.e. | | | | 0.11 | | 0.09 | 0.13 | | | |
| | P-value | | | | **0.01** | | **0.00** | **0.08** | | | |
| | Odds Ratio | | | | 0.76 | | 2.30 | 1.24 | | | |
| Belgium Flemish | Coefficient | -0.61 | 0.71 | 0.19 | | | 0.48 | | | | |
| | Coeff_s.e. | 0.21 | 0.38 | 0.11 | | | 0.10 | | | | |
| | P-value | **0.01** | **0.06** | **0.09** | | | **0.00** | | | | |
| | Odds Ratio | 0.54 | 2.03 | 1.21 | | | 1.62 | | | | |
| Belgium French | Coefficient | | | | | | 0.42 | 0.62 | | | 0.30 |
| | Coeff_s.e. | | | | | | 0.19 | 0.18 | | | 0.15 |
| | P-value | | | | | | **0.03** | **0.00** | | | **0.05** |
| | Odds Ratio | | | | | | 1.52 | 1.85 | | | 1.35 |
| Bulgaria | Coefficient | | | -0.22 | 0.30 | -0.24 | 0.48 | | 0.15 | | |
| | Coeff_s.e. | | | 0.11 | 0.09 | 0.09 | 0.11 | | 0.09 | | |
| | P-value | | | **0.05** | **0.00** | **0.01** | **0.00** | | **0.09** | | |
| | Odds Ratio | | | 0.80 | 1.34 | 0.79 | 1.62 | | 1.16 | | |
| Czech Republic | Coefficient | | 0.80 | 0.27 | -0.38 | | 0.66 | | | | 0.21 |
| | Coeff_s.e. | | 0.28 | 0.11 | 0.10 | | 0.10 | | | | 0.09 |
| | P-value | | **0.01** | **0.02** | **0.00** | | **0.00** | | | | **0.02** |
| | Odds Ratio | | 2.23 | 1.31 | 0.69 | | 1.94 | | | | 1.23 |
| Denmark | Coefficient | | 1.71 | 0.18 | | | 0.76 | | 0.31 | | |
| | Coeff_s.e. | | 0.75 | 0.09 | | | 0.09 | | 0.12 | | |
| | P-value | | **0.02** | **0.06** | | | **0.00** | | **0.01** | | |
| | Odds Ratio | | 5.51 | 1.19 | | | 2.14 | | 1.36 | | |
| Finland | Coefficient | | | | | 0.30 | 0.65 | | 0.53 | | |
| | Coeff_s.e. | | | | | 0.11 | 0.11 | | 0.08 | | |
| | P-value | | | | | **0.01** | **0.00** | | **0.00** | | |
| | Odds Ratio | | | | | 1.35 | 1.91 | | 1.70 | | |
| France | Coefficient | | | | | | 0.68 | | 0.57 | | |
| | Coeff_s.e. | | | | | | 0.09 | | 0.13 | | |
| | P-value | | | | | | **0.00** | | **0.00** | | |
| | Odds Ratio | | | | | | 1.97 | | 1.78 | | |
| Germany | Coefficient | | 0.31 | | -0.23 | | 0.54 | 0.22 | | | 0.23 |
| | Coeff_s.e. | | 0.19 | | 0.11 | | 0.09 | 0.13 | | | 0.12 |
| | P-value | | **0.09** | | **0.04** | | **0.00** | **0.09** | | | **0.07** |
| | Odds Ratio | | 1.37 | | 0.79 | | 1.71 | 1.25 | | | 1.26 |
| Hungary | Coefficient | | | 0.25 | | | 0.79 | | | -0.18 | |
| | Coeff_s.e. | | | 0.09 | | | 0.09 | | | 0.11 | |
| | P-value | | | **0.01** | | | **0.00** | | | **0.10** | |
| | Odds Ratio | | | 1.28 | | | 2.21 | | | 0.84 | |
| Ireland | Coefficient | | | 0.36 | -0.26 | | 0.59 | | 0.68 | | |
| | Coeff_s.e. | | | 0.11 | 0.10 | | 0.09 | | 0.12 | | |
| | P-value | | | **0.00** | **0.01** | | **0.00** | | **0.00** | | |
| | Odds Ratio | | | 1.44 | 0.77 | | 1.81 | | 1.97 | | |
| Italy | Coefficient | | | 0.19 | | -0.24 | 0.71 | | | 0.16 | |
| | Coeff_s.e. | | | 0.11 | | 0.12 | 0.11 | | | 0.09 | |
| | P-value | | | **0.08** | | **0.05** | **0.00** | | | **0.07** | |
| | Odds Ratio | | | 1.21 | | 0.79 | 2.04 | | | 1.17 | |
| Latvia | Coefficient | -0.57 | | 0.22 | -0.32 | | 0.78 | | 0.37 | | |
| | Coeff_s.e. | 0.20 | | 0.13 | 0.16 | | 0.12 | | 0.13 | | |
| | P-value | **0.01** | | **0.09** | **0.05** | | **0.00** | | **0.01** | | |
| | Odds Ratio | 0.57 | | 1.24 | 0.72 | | 2.19 | | 1.44 | | |
| Lithuania | Coefficient | -0.38 | | | | -0.25 | 0.68 | | 0.70 | | |
| | Coeff_s.e. | 0.22 | | | | 0.10 | 0.13 | | 0.14 | | |
| | P-value | **0.08** | | | | **0.02** | **0.00** | | **0.00** | | |
| | Odds Ratio | 0.68 | | | | 0.78 | 1.97 | | 2.01 | | |

(*Continued*)

**Table 5.** (Continued)

| | | Gender | Preprimary | Sensebelonging | Engagedreading | Likereading | Confident | LitActivities | LitTask | ParentsPercep | ParentsLikeRead |
|---|---|---|---|---|---|---|---|---|---|---|---|
| Malta | Coefficient | -0.39 | | | 0.27 | | 0.82 | | 0.23 | | |
| | Coeff_s.e. | 0.24 | | | 0.12 | | 0.12 | | 0.14 | | |
| | P-value | **0.10** | | | **0.03** | | **0.00** | | **0.09** | | |
| | Odds Ratio | 0.68 | | | 1.31 | | 2.27 | | 1.26 | | |
| Netherlands | Coefficient | | | 0.27 | -0.53 | 0.61 | 0.50 | | 0.34 | | 0.29 |
| | Coeff_s.e. | | | 0.13 | 0.17 | 0.15 | 0.09 | | 0.15 | | 0.13 |
| | P-value | | | **0.03** | **0.00** | **0.00** | **0.00** | | **0.02** | | **0.03** |
| | Odds Ratio | | | 1.31 | 0.59 | 1.83 | 1.65 | | 1.40 | | 1.34 |
| Northen Ireland | Coefficient | | | 0.19 | -0.41 | 0.31 | 0.78 | | | | |
| | Coeff_s.e. | | | 0.11 | 0.12 | 0.11 | 0.09 | | | | |
| | P-value | | | **0.09** | **0.00** | **0.00** | **0.00** | | | | |
| | Odds Ratio | | | 1.21 | 0.67 | 1.36 | 2.19 | | | | |
| Poland | Coefficient | -0.58 | | | | -0.28 | 0.61 | -0.26 | 0.57 | | 0.28 |
| | Coeff_s.e. | 0.18 | | | | 0.11 | 0.09 | 0.10 | 0.14 | | 0.11 |
| | P-value | **0.00** | | | | **0.01** | **0.00** | **0.01** | **0.00** | | **0.01** |
| | Odds Ratio | 0.56 | | | | 0.76 | 1.84 | 0.77 | 1.77 | | 1.32 |
| Portugal | Coefficient | | | | | -0.23 | 0.83 | | | | |
| | Coeff_s.e. | | | | | 0.07 | 0.09 | | | | |
| | P-value | | | | | **0.00** | **0.00** | | | | |
| | Odds Ratio | | | | | 0.80 | 2.29 | | | | |
| Slovak Republic | Coefficient | | 0.83 | | | | 0.60 | | | | 0.17 |
| | Coeff_s.e. | | 0.24 | | | | 0.07 | | | | 0.09 |
| | P-value | | **0.00** | | | | **0.00** | | | | **0.06** |
| | Odds Ratio | | 2.30 | | | | 1.83 | | | | 1.18 |
| Slovenia | Coefficient | -0.70 | | 0.29 | -0.50 | | 0.74 | 0.29 | | -0.20 | |
| | Coeff_s.e. | 0.21 | | 0.12 | 0.13 | | 0.10 | 0.10 | | 0.11 | |
| | P-value | **0.00** | | **0.02** | **0.00** | | **0.00** | **0.00** | | **0.06** | |
| | Odds Ratio | 0.49 | | 1.33 | 0.61 | | 2.09 | 1.34 | | 0.82 | |
| Spain | Coefficient | | | 0.15 | | -0.15 | 0.60 | 0.17 | 0.32 | | |
| | Coeff_s.e. | | | 0.08 | | 0.06 | 0.08 | 0.08 | 0.08 | | |
| | P-value | | | **0.06** | | **0.01** | **0.00** | **0.03** | **0.00** | | |
| | Odds Ratio | | | 1.17 | | 0.86 | 1.82 | 1.18 | 1.38 | | |
| Sweden | Coefficient | | 0.86 | | -0.34 | | 0.62 | -0.20 | 0.61 | | |
| | Coeff_s.e. | | 0.33 | | 0.13 | | 0.12 | 0.10 | 0.13 | | |
| | P-value | | **0.01** | | **0.01** | | **0.00** | **0.05** | **0.00** | | |
| | Odds Ratio | | 2.37 | | 0.71 | | 1.85 | 0.82 | 1.84 | | |

divergent results, with different effects from one country to another, and something similar occurred with *reading for pleasure*.

For the family-related variables, we found that in most countries (18 out of 23), at least one of the following variables was statistically significant in the prediction of academic resilience: having done early literacy activities in the family environment, having done literacy activities before starting schooling, and parents' reading for pleasure. The first of these was significant in more than half of the countries.

In general, the results with regard to teaching-related variables indicated a smaller predictive capacity for academic resilience than those related to students or their family characteristics. The two variables demonstrating greatest predictive capacity were *a safe and orderly school environment* and *co-existence in schools*, confirming the results reported by Erberber, et al. [17]. There was a mix of instructional type variables which accumulated positive effects in a notable number of countries, particularly classroom work being oriented towards achieving academic objectives and some characteristics of teaching practices. In this regard, teaching

**Table 6. Coefficient, p-value, and odds ratio of variables related to teaching.**

| | | Gender | BasicTraining | CompleTraining | Emphasis | Security | Interaction | Satisfaction | Limitation | Rutinare | Sistemat | Comprehensive | Reflective | Homework | Readings |
|---|---|---|---|---|---|---|---|---|---|---|---|---|---|---|---|
| Austria | Coefficient | | | | 0.50 | | | | | | -0.21 | | | | |
| | Coeff_s.e. | | | | 0.16 | | | | | | 0.12 | | | | |
| | P-value | | | | **0.00** | | | | | | **0.10** | | | | |
| | Odds Ratio | | | | 1.64 | | | | | | 0.81 | | | | |
| Belgium Flemish | Coefficient | | | | | 0.27 | | | 0.21 | | | -0.23 | 0.27 | -0.20 | |
| | Coeff_s.e. | | | | | 0.12 | | | 0.12 | | | 0.12 | 0.16 | 0.10 | |
| | P-value | | | | | **0.02** | | | **0.08** | | | **0.05** | **0.09** | **0.04** | |
| | Odds Ratio | | | | | 1.31 | | | 1.24 | | | 0.79 | 1.31 | 0.82 | |
| Belgium French | Coefficient | | | 0.33 | 0.62 | | | | | | | | 0.25 | | |
| | Coeff_s.e. | | | 0.19 | 0.21 | | | | | | | | 0.14 | | |
| | P-value | | | **0.08** | **0.00** | | | | | | | | **0.08** | | |
| | Odds Ratio | | | 1.39 | 1.86 | | | | | | | | 1.28 | | |
| Bulgaria | Coefficient | | | | | 0.41 | | -0.28 | | | | | 0.48 | | -0.34 |
| | Coeff_s.e. | | | | | 0.18 | | 0.17 | | | | | 0.21 | | 0.16 |
| | P-value | | | | | **0.02** | | **0.10** | | | | | **0.03** | | **0.03** |
| | Odds Ratio | | | | | 1.50 | | 0.75 | | | | | 1.61 | | 0.71 |
| Czech Republic | Coefficient | | | | | | | 0.27 | | | | | | | -0.18 |
| | Coeff_s.e. | | | | | | | 0.12 | | | | | | | 0.10 |
| | P-value | | | | | | | **0.02** | | | | | | | **0.09** |
| | Odds Ratio | | | | | | | 1.31 | | | | | | | 0.84 |
| Denmark | Coefficient | | | -0.18 | 0.24 | | -0.28 | | | | | | 0.17 | | |
| | Coeff_s.e. | | | 0.11 | 0.14 | | 0.15 | | | | | | 0.09 | | |
| | P-value | | | **0.10** | **0.08** | | **0.08** | | | | | | **0.08** | | |
| | Odds Ratio | | | 0.83 | 1.28 | | 0.76 | | | | | | 1.18 | | |
| Finland | Coefficient | | | -0.27 | | | | -0.13 | 0.22 | | | | | | |
| | Coeff_s.e. | | | 0.09 | | | | 0.08 | 0.13 | | | | | | |
| | P-value | | | **0.01** | | | | **0.10** | **0.10** | | | | | | |
| | Odds Ratio | | | 0.77 | | | | 0.88 | 1.24 | | | | | | |
| France | Coefficient | | | | | 0.42 | -0.31 | 0.20 | | | | | | | 0.24 |
| | Coeff_s.e. | | | | | 0.12 | 0.12 | 0.12 | | | | | | | 0.11 |
| | P-value | | | | | **0.00** | **0.01** | **0.10** | | | | | | | **0.04** |
| | Odds Ratio | | | | | 1.52 | 0.74 | 1.22 | | | | | | | 1.27 |
| Germany | Coefficient | | | | 0.30 | 0.27 | | | | | | | 0.28 | | |
| | Coeff_s.e. | | | | 0.19 | 0.14 | | | | | | | 0.13 | | |
| | P-value | | | | **0.11** | **0.06** | | | | | | | **0.04** | | |
| | Odds Ratio | | | | 1.35 | 1.31 | | | | | | | 1.32 | | |
| Hungary | Coefficient | | | | | | | | 0.31 | | | | | | |
| | Coeff_s.e. | | | | | | | | 0.14 | | | | | | |
| | P-value | | | | | | | | **0.03** | | | | | | |
| | Odds Ratio | | | | | | | | 1.36 | | | | | | |
| Ireland | Coefficient | -0.40 | | | | 0.15 | | | | | -0.28 | | 0.18 | | |
| | Coeff_s.e. | 0.23 | | | | 0.08 | | | | | 0.13 | | 0.09 | | |
| | P-value | **0.08** | | | | **0.05** | | | | | **0.04** | | **0.05** | | |
| | Odds Ratio | 0.67 | | | | 1.17 | | | | | 0.76 | | 1.20 | | |
| Italy | Coefficient | -1.09 | | 0.23 | | 0.48 | -0.25 | -0.21 | | | | | 0.45 | | -0.22 |
| | Coeff_s.e. | 0.41 | | 0.07 | | 0.15 | 0.10 | 0.11 | | | | | 0.15 | | 0.11 |
| | P-value | **0.01** | | **0.00** | | **0.00** | **0.02** | **0.06** | | | | | **0.00** | | **0.06** |
| | Odds Ratio | 0.34 | | 1.26 | | 1.62 | 0.78 | 0.81 | | | | | 1.57 | | 0.80 |

*(Continued)*

**Table 6.** (Continued)

| | | Gender | BasicTraining | CompleTraining | Emphasis | Security | Interaction | Satisfaction | Limitation | Rutinare | Sistemat | Comprehensive | Reflective | Homework | Readings |
|---|---|---|---|---|---|---|---|---|---|---|---|---|---|---|---|
| Latvia | Coefficient | | | | | | | | | | | | 0.29 | | |
| | Coeff_s.e. | | | | | | | | | | | | 0.17 | | |
| | P-value | | | | | | | | | | | | **0.10** | | |
| | Odds Ratio | | | | | | | | | | | | 1.34 | | |
| Lithuania | Coefficient | | | | | | | | | | 0.46 | | | | |
| | Coeff_s.e. | | | | | | | | | | 0.24 | | | | |
| | P-value | | | | | | | | | | **0.06** | | | | |
| | Odds Ratio | | | | | | | | | | 1.58 | | | | |
| Malta | Coefficient | | -0.39 | 0.33 | 0.32 | | -0.33 | | | | | | | | |
| | Coeff_s.e. | | 0.19 | 0.15 | 0.13 | | 0.18 | | | | | | | | |
| | P-value | | **0.04** | **0.03** | **0.01** | | **0.06** | | | | | | | | |
| | Odds Ratio | | 0.67 | 1.40 | 1.37 | | 0.72 | | | | | | | | |
| Netherlands | Coefficient | | | | 0.34 | 0.33 | | | | | | 0.33 | -0.47 | | |
| | Coeff_s.e. | | | | 0.17 | 0.16 | | | | | | 0.18 | 0.22 | | |
| | P-value | | | | **0.04** | **0.04** | | | | | | **0.06** | **0.03** | | |
| | Odds Ratio | | | | 1.41 | 1.39 | | | | | | 1.40 | 0.63 | | |
| Northen Ireland | Coefficient | | | 0.20 | -0.20 | | 0.21 | | 0.38 | 0.30 | -0.28 | | | | |
| | Coeff_s.e. | | | 0.12 | 0.12 | | 0.11 | | 0.13 | 0.15 | 0.15 | | | | |
| | P-value | | | **0.10** | **0.10** | | **0.06** | | **0.01** | **0.05** | **0.07** | | | | |
| | Odds Ratio | | | 1.22 | 0.82 | | 1.24 | | 1.46 | 1.35 | 0.76 | | | | |
| Poland | Coefficient | | | | | -0.21 | | | | | | | | | |
| | Coeff_s.e. | | | | | 0.11 | | | | | | | | | |
| | P-value | | | | | **0.06** | | | | | | | | | |
| | Odds Ratio | | | | | 0.81 | | | | | | | | | |
| Portugal | Coefficient | | | 0.16 | | 0.18 | | | 0.24 | | | | | | |
| | Coeff_s.e. | | | 0.09 | | 0.09 | | | 0.10 | | | | | | |
| | P-value | | | **0.07** | | **0.05** | | | **0.01** | | | | | | |
| | Odds Ratio | | | 1.17 | | 1.19 | | | 1.27 | | | | | | |
| Slovak Republic | Coefficient | -0.56 | | | 0.30 | | | | | | | 0.30 | | | |
| | Coeff_s.e. | 0.27 | | | 0.14 | | | | | | | 0.17 | | | |
| | P-value | **0.04** | | | **0.03** | | | | | | | **0.09** | | | |
| | Odds Ratio | 0.57 | | | 1.35 | | | | | | | 1.35 | | | |
| Slovenia | Coefficient | | | | | | -0.28 | | | | | | 0.43 | | |
| | Coeff_s.e. | | | | | | 0.13 | | | | | | 0.13 | | |
| | P-value | | | | | | **0.03** | | | | | | **0.00** | | |
| | Odds Ratio | | | | | | 0.76 | | | | | | 1.53 | | |
| Spain | Coefficient | | | | | | | | 0.12 | -0.28 | | 0.16 | | | |
| | Coeff_s.e. | | | | | | | | 0.08 | 0.07 | | 0.07 | | | |
| | P-value | | | | | | | | **0.10** | **0.00** | | **0.03** | | | |
| | Odds Ratio | | | | | | | | 1.13 | 0.76 | | 1.18 | | | |
| Sweden | Coefficient | | -0.25 | 0.33 | | | | | 0.49 | | | | | 0.24 | |
| | Coeff_s.e. | | 0.15 | 0.13 | | | | | 0.12 | | | | | 0.13 | |
| | P-value | | **0.10** | **0.01** | | | | | **0.00** | | | | | **0.06** | |
| | Odds Ratio | | 0.78 | 1.39 | | | | | 1.63 | | | | | 1.27 | |

**Table 7. Percentage of countries where the coefficient was significant for student and family contextual variables.**

| Variable | Percentage |
|---|---|
| Gender | 26.1% |
| Preprimary | 26.1% |
| Sensebelonging | 52.2% |
| Engagedreading | 47.8% |
| Likereading | 43.5% |
| Confident | 100.0% |
| LitActivities | 26.1% |
| LitTask | 52.2% |
| ParentsPercep | 13.0% |
| ParentsLikeRead | 26.1% |

which more often used comprehension and reflective reading techniques predicted greater likelihood of resilience than teaching practices for reading based on routine, systematic, repetitive procedures. These results are in line with previous evidence indicating the importance of teaching practices in school performance [33, 40–43, 45, 46]. Finally, it is worth noting that in more than a quarter of the countries there was a positive, significant relationship between academic resilience and teacher participation in complementary training activities in areas such as reading theory, corrective reading, and reading evaluation methods. These results are in line with previous studies which have highlighted the role of continued teacher training in improving students' learning outcomes [51–53].

In terms of the weight of the predictor variables, we saw that the confidence in reading index increased the probability of student resilience between 62 percentage points (Belgium–Flemish community) and 130 percentage points (Austria and Poland). Students who had a strong sense of belonging to the school they attended generally had a better chance of being resilient than those who did not, up to 40 percentage points higher, as in Ireland. Being a boy was associated with lower likelihood of resilience, up to 50% lower in the case of Slovenia. Early literacy activities in the family setting increased the chances of being academically resilient, in Lithuania for example, students in unfavorable socioeconomic situations who had

**Table 8. Percentage of countries where the coefficient was significant for variables related to teaching.**

| Variable | Percentage |
|---|---|
| Gender | 13.0% |
| BasicTraining | 8.7% |
| CompleTraining | 34.8% |
| Emphasis | 34.8% |
| Security | 39.1% |
| Interaction | 26.1% |
| Satisfaction | 17.4% |
| Limitation | 34.8% |
| Rutinare | 13.0% |
| Sistemat | 13.0% |
| Comprehensive | 30.4% |
| Reflective | 30.4% |
| Homework | 8.7% |
| Readings | 17.4% |

done these kinds of activities were twice as likely to be resilient than those who had low scores in this index. Similarly, as in Denmark, attending preprimary school multiplied the chances of being resilient by 5.5. With regard to the teacher-related variables, we estimate that working in a safe environment can increase the likelihood of students being resilient by up to 62 percentage points, as we saw in Italy. In addition, students whose teachers had received complementary training, understood as improvements in understanding the theory of reading, reading evaluation, and corrective reading procedures, were 40 percentage points more likely to be resilient (Malta). Students attending schools with a strong academic emphasis were almost twice as likely to be resilient (Belgium–French-speaking community). Teachers who reported feeling limited by the characteristics of their students were associated with between 13 and 64 percentage points increase in the probability of students being resilient. Teaching practices associated with reading comprehension and reflective reading were linked to increases in the probability of resilience of up to 57 points (Italy) and 61 points (Bulgaria), which confirms the results from Lavy [44] and Rjosk et al. [47].

In summary, educational measures aimed at increasing student confidence, studying in safe environments that increase a sense of school belonging, and a strong academic emphasis, increase students' academic resilience. Good initial training which allows teachers to deliver teaching practices that encourage student learning will increase the number of resilient students, and consequently will improve the education system. It is worth noting a limitation of our study, which is that there are variables that we did not evaluate in this study but which may be important when predicting and explaining academic resilience. These include *grit*, *emotional intelligence*, and other non-cognitive variables [77, 78]. Similarly, there are variables that were not collected in this study, such as IQ or a background of mental issues (mild intellectual disabilities that do not stop the students from taking the test), which might have an impact on the capacity for resilience, as well as the fact of having repeated a school year or not, which is indicative of prior performance. Nor should it be forgotten that context questionnaires on very many occasions refer to a respondents' perceptions which may not correspond to the reality of the social surroundings, responses may be associated with social desirability, or socio-cultural perceptions that are deep-rooted in the students' environment.

## Author Contributions

**Conceptualization:** Francisco J. García-Crespo, Rubén Fernández-Alonso, José Muñiz.

**Data curation:** Francisco J. García-Crespo.

**Formal analysis:** Francisco J. García-Crespo.

**Funding acquisition:** Rubén Fernández-Alonso, José Muñiz.

**Methodology:** Francisco J. García-Crespo, Rubén Fernández-Alonso, José Muñiz.

**Supervision:** Rubén Fernández-Alonso, José Muñiz.

**Writing – original draft:** Francisco J. García-Crespo.

**Writing – review & editing:** Francisco J. García-Crespo, Rubén Fernández-Alonso, José Muñiz.

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
