## [Decision Letter · Decision Letter 0]

11 Feb 2021

PONE-D-21-00014

Academic Resilience in European countries: The role of Teachers, Families, and Student profiles

PLOS ONE

Dear Dr. R.Alonso,

Thank you for submitting your manuscript to PLOS ONE. After careful consideration, we feel that it has merit but does not fully meet PLOS ONE’s publication criteria as it currently stands. Therefore, we invite you to submit a revised version of the manuscript that addresses the points raised during the review process.

We look forward to receiving your revised manuscript.

Kind regards,

Eduardo Fonseca-Pedrero, PhD

Academic Editor

PLOS ONE

Additional Editor Comments (if provided):

The work entitled "Academic Resilience in European countries: The role of Teachers, Families, and Student profiles ” is a relevant research paper within the educational field. I have a few comments to make that should be addressed before I recommend this manuscript for publication in PLOS ONE:

1.- Abstract: please add more (empirical) information in results section.

2.- Add more information about the sampling procedure and sample (sociodemographic variables), in particular about inclusion/exclusion criteria.

3.- Do you have any information about non-response? Describe Were outliers removed from the data? Which method did you use to deal with missing data in the analyses?

4.- Add information about the psychometric properties of the measurement instruments used. (e.g., measrument invariance issues across countries)

5- How do you control other potential covariables? (IQ, socio-economic status, previous history of mental disorders, etc.)

6. Please, add some limitations of the present study (e.g., self-reports, cross-cultural issues).

Journal Requirements:

Reviewers' comments:

Reviewer's Responses to Questions

**Comments to the Author**

1. Is the manuscript technically sound, and do the data support the conclusions?

Reviewer #1: Yes

Reviewer #2: Yes

2. Has the statistical analysis been performed appropriately and rigorously? 

Reviewer #1: Yes

Reviewer #2: Yes

3. Have the authors made all data underlying the findings in their manuscript fully available?

Reviewer #1: Yes

Reviewer #2: Yes

4. Is the manuscript presented in an intelligible fashion and written in standard English?

Reviewer #1: Yes

Reviewer #2: Yes

5. Review Comments to the Author

Reviewer #1: Dear authors,

Thank you very much for your contribution. I have enjoyed reading this paper very much. I particularly appreciate the robustness of the data used and the multilevel approach to provide meaningful interpretations of the results. The introduction is also very clear signaling the importance of identifying potential moderators of academic resilience across countries.

As pointed out by the authors, the results are aligned with expectations. It is indeed not surprising that confidence in reading was the variable most strongly linked to academic resilience. Self-perception of competence and reading performance are mutually reinforcing. So when higher-performing students receive and process performance feedback, their perception of competence tends to be higher. Since academic resilient students are defined as students in the top quartile of reading comprehension, it would be reasonable to expect that they would have higher confidence in reading. It would be good to explain this association further when discussing the results (see Marsh and Craven, (2006)).

Furthermore, this would likely impact most if not all, student-level variables. To some extent, it may explain why student-level variables are comparatively stronger in magnitude than teacher-level variables. Therefore, I would suggest considering to use grade repetition (or past grades if available) as a proxy to control for past achievement. Otherwise, it might good to point this out in the limitations of the study. The consequences of comparing the strengths of students and family variables with variables related to teaching are important from a policy perspective, as students and family variables are usually under a softer influence from education policies than teaching practices.

Finally, I would suggest considering to add summary tables/figures to help to visualize the results. For example, the average beta/odds ratio (and SD) per variable, percentage of countries where the coefficient was significant, EU average results, etc. I would also suggest considering adding (or citing) the percentage of resilient students by countries and the correlation matrix of the models' variables.

Marsh, H. and R. Craven (2006), “Reciprocal Effects of Self-Concept and Performance From a Multidimensional Perspective: Beyond Seductive Pleasure and Unidimensional Perspectives”, Perspectives on Psychological Science, Vol. 1/2, pp. 133-163, http://dx.doi.org/10.1111/j.1745-6916.2006.00010.x.

Reviewer #2: The paper “Academic Resilience in European countries: The role of Teachers, Families, and Student profiles” addresses a relevant topic which contributed to improve educational policies. I think the paper is well-written and clear, although I missed the authors described more details about the results. I just have some minor comments.

The literature review clearly exposes the relevance of the study based on previous researches. However, there is just a short reference to the influence of teachers´ training, and it is relevant to understand why the authors include them in the model and also to help in the discussion of the results. I would add some information about how the teachers´ training could affect the students´ academic resilience. It is clear why the teachers´ variables related to academic practices are included, but how is the training related to resilience? More evidence needed.

In table 1, there is a column for the number of home questionnaires. It is not clear why there is not a home questionnaire for all the students. It should be clarified in page 7 (around line 119) in order to help understanding the data.

In page 9, the authors describe all the variables used. However, there are some where the details are not complete, such as “engaged in reading lessons”, “like reading” and “early literacy activities before school”. It also happens with some variables related to family and teachers. I think the information about the variables should be consistent. Either including all the variables or not doing that, but doing the same thing in all the cases. It could be easier to include a table where all the variables are deeply explained. It would make that part less repetitive.

After tables 2 and 3 the readers would expect a description of the results. All the results are included in the discussion section but it would be illustrative to summarize main results in the results section.

In the discussion section, I missed a discussion about potential patterns found between countries. The authors point to cultural and social elements as responsible of some of the results but: are similar countries reaching the same results? Are countries with similar characteristics having the same variables affecting the academic resilience?

In relation to the teachers training, is it related to the resilience because the topics of the training are directly related to the topic measured as performance in students?

Other minor issues:

- Page 14, line 272: the expression “may vey well be biased” is not clear to me.

- Page 15, line 287- significance level should be at 1% instead of at 10%.

- In table 2 you use “sex” and in table 3 you use “gender”. Maybe you can unify the use of the term as I assume you are referring to the same thing in both cases.

6. PLOS authors have the option to publish the peer review history of their article (what does this mean?). If published, this will include your full peer review and any attached files.

Reviewer #1: No

Reviewer #2: No

---

## [Author Response · Author response to Decision Letter 0]

1 Jun 2021

Response to reviewers is sent as an attachment

---

## [Editor Report · Decision Letter 1]

7 Jun 2021

Academic Resilience in European countries: The role of Teachers, Families, and Student profiles

PONE-D-21-00014R1

Dear Dr. Rubén Fernández,

We’re pleased to inform you that your manuscript has been judged scientifically suitable for publication and will be formally accepted for publication once it meets all outstanding technical requirements.

Kind regards,

Eduardo Fonseca-Pedrero, PhD

Academic Editor

PLOS ONE
---

## [Editor Report · Acceptance letter]

25 Jun 2021

PONE-D-21-00014R1 

Academic resilience in European countries: The role of teachers, families, and student profiles 

Dear Dr. Fernández-Alonso:

I'm pleased to inform you that your manuscript has been deemed suitable for publication in PLOS ONE. Congratulations! Your manuscript is now with our production department. 

Kind regards, 

on behalf of

Dr. Eduardo Fonseca-Pedrero 

Academic Editor

PLOS ONE